

# Linear growth of the entanglement entropy
# for quadratic Hamiltonians and arbitrary initial states

## Giacomo De Palma[1*] and Lucas Hackl[2†]

**1** Scuola Normale Superiore, 56126 Pisa, Italy
**2** School of Mathematics and Statistics & School of Physics,
The University of Melbourne, Parkville, VIC 3010, Australia

⋆ giacomo.depalma@sns.it, † lucas.hackl@unimelb.edu.au

## Abstract

We prove that the entanglement entropy of any pure initial state of a bipartite bosonic quantum system grows linearly in time with respect to the dynamics induced by any unstable quadratic Hamiltonian. The growth rate does not depend on the initial state and is equal to the sum of certain Lyapunov exponents of the corresponding classical dynamics. This paper generalizes the findings of [Bianchi *et al.*, JHEP 2018, 25 (2018)], which proves the same result in the special case of Gaussian initial states. Our proof is based on a recent generalization of the strong subadditivity of the von Neumann entropy for bosonic quantum systems [De Palma *et al.*, arXiv:2105.05627]. This technique allows us to extend our result to generic mixed initial states, with the squashed entanglement providing the right generalization of the entanglement entropy. We discuss several applications of our results to physical systems with (weakly) interacting Hamiltonians and periodically driven quantum systems, including certain quantum field theory models.

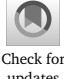

# 1   Introduction

Entanglement is a cornerstone of quantum theory and its dynamics has been extensively studied in a wide range of different systems [1–5]. It also provides an important link between classical chaos and quantum chaos in the context of Lyapunov instabilities [6]. The most prominent entanglement measure for pure states (*i.e.*, rank-one projectors) is the entanglement entropy [7, 8]. The entanglement entropy of the pure state $\rho$ of the bipartite quantum system $AB$[1] is defined as

$$S(A)(\rho) = -\text{Tr}[\rho_A \ln \rho_A],  \tag{1}$$

where $S$ denotes the von Neumann entropy [9–11]. Saturation of the entanglement entropy is considered as signature of thermalization or equilibration. The transition from an initially linear growth to such an eventual saturation has been also studied in the context of quenches of both integrable and non-integrable systems [12, 13]. More recently, out-of-time-order correlators have been used as valuable tool to describe thermodynamic processes and to define quantum Lyapunov exponents [14, 15].

To our knowledge, the connection between linear growth of the entanglement entropy and classical Lyapunov exponents was first observed by Asplund and Berenstein in [16] for a stroboscobic Hamiltonian coupling two bosonic modes. They found that the entanglement entropy grew as the sum over the positive Lyapunov exponents and already conjectured how this finding should apply more generally. This conjecture was proven for the case of time-dependent quadratic Hamiltonians with Gaussian initial states in [17, 18]. In particular, this lead to an algorithm [18] for determining which Lyapunov exponents need to be summed for any chosen subsystem and it showed under what conditions the growth rate agreed with the famous classical Kolmogorov–Sinai entropy rate. More recently, Modak, Rigol, Bianchi and one of the present authors gave numerical evidence [19] that the same growth rates also apply to non-Gaussian initial states and identified a subleading logarithmic correction for a certain class of "meta-stable" Hamiltonians. Linear growth was also observed in a toy model for time evolving QFT with controllable chaos [20].

On a technical level, the main proof [18] for linear growth for Gaussian initial states was based on relating the asymptotic growth of the entanglement entropy

$$S(A)(t) \sim \ln \text{Vol} \, \mathcal{V}_A(t)  \tag{2}$$

to the volume of a time-dependent parallelepiped $\mathcal{V}_A(t)$ in the dual phase space. The resulting asymptotics applies to arbitrary pure Gaussian states and also serves as an upper bound for arbitrary initial states with finite covariance matrix, as Gaussian states have maximal entropy among all states with given covariance matrix. Proving that the respective growth rate applies

---

[1]A *quantum system A* is given by a Hilbert space $\mathcal{H}_A$. A *quantum state* of $A$ is a positive semidefinite linear operator with unit trace acting on $\mathcal{H}_A$. Given two quantum systems $A$ and $B$ with Hilbert spaces $\mathcal{H}_A$ and $\mathcal{H}_B$, respectively, their union is the bipartite quantum system $AB$ with Hilbert space $\mathcal{H}_{AB} = \mathcal{H}_A \otimes \mathcal{H}_B$. Given a quantum state $\rho$ of $AB$, its *marginal state* on $A$ is $\rho_A = \text{Tr}_B \rho$, where $\text{Tr}_B$ denotes the partial trace over $\mathcal{H}_B$.

to arbitrary (non-Gaussian) initial states requires us to bound the entanglement entropy from below, which is in general a very difficult problem. A recent work [21] by Trevisan and one of the present authors introduced a novel relation between the mutual information of Gaussian and non-Gaussian states. More precisely, let $AB$ be a bipartite bosonic quantum system, let $M$ be a symplectic transformation[2] and let $U_M$ be the unitary operator that implements $M$ in the Hilbert space of $AB$, *i.e.*, $U_M$ implements the linear transformation of the quadratures given by $M$. Then, for any (generically mixed) state $\rho$ of $AB$ with finite covariance matrix, we have

$$I(A;B)(\rho) + I(A;B)(\mathcal{U}_M(\rho)) \geq \inf_{\sigma \text{ Gaussian}} \left( I(A;B)(\sigma) + I(A;B)(\mathcal{U}_M(\rho)) \right), \tag{3}$$

where

$$I(A;B)(\rho) = S(A)(\rho) + S(B)(\rho) - S(AB)(\rho) \tag{4}$$

is the mutual information of $\rho$ across the subsystems $A$ and $B$ [9–11] and $\mathcal{U}_M$ is the quantum channel associated to $U_M$, *i.e.*,

$$\mathcal{U}_M(\rho) = U_M \, \rho \, U_M^\dagger. \tag{5}$$

The left-hand side of (3) is bounded from below by a minimization over (generally mixed) Gaussian states $\sigma$. To use this result for bounding the growth of the entanglement entropy, we need to consider a time-dependent symplectic transformation $M(t)$ describing the classical evolution induced by a quadratic Hamiltonian and determine the growth of the right-hand side in (3). While it is easy to show that the right-hand side will generally grow linearly in time for a fixed Gaussian state $\sigma$, it turns out to be non-trivial to show that taking the time asymptotics for fixed $\sigma$ commutes with taking the infimum over $\sigma$ for fixed time $t$. In [21], this was only done for the special class of time-independent Hamiltonians giving rise positive-definite symplectic transformations $M(t)$, for which a linear growth with undetermined coefficient was found.

The present paper combines these recent findings of [21] with the insights about the entanglement entropy growth for Gaussian states of [18]. This allows us to treat the most general case of an arbitrary time-dependent quadratic Hamiltonian $\hat{H}(t)$ (with well-defined Lyapunov exponents) and an arbitrary pure initial state $\rho$ of a bipartite bosonic quantum system $AB$, for which we prove that the entanglement entropy grows as

$$S(A)(t) = \Lambda_A \, t + o(t) \quad \text{as} \quad t \to \infty, \tag{6}$$

where $\Lambda_A = \sum_{j=1}^{2N_A} \lambda_{i_i}$ is the subsystem coefficient associated to $A$ computed as a sum over $2N_A$ Lyapunov exponents $\lambda_i$ according to algorithm from [18], and $N_A$ is the number of modes of $A$.

We also extend this result to mixed states, where the entanglement entropy is replaced by the squashed entanglement, also called CMI entanglement [22–31]. Let $\rho$ be a state of the bipartite quantum system $AB$ such that both $S(A)(\rho)$ and $S(B)(\rho)$ are finite. The squashed entanglement of $\rho$ is defined as the following infimum over all the possible extensions $\tilde{\rho}$ of $\rho$ on a tripartite quantum system $ABR$, where $R$ is an arbitrary finite-dimensional quantum system[3]:

$$E_{\text{sq}}(\rho) = \frac{1}{2} \inf \left\{ I(A;B|R)(\tilde{\rho}) : \text{Tr}_R \tilde{\rho} = \rho, \, \dim \mathcal{H}_R < \infty \right\}. \tag{7}$$

Here

$$I(A;B|R) = S(A|R) + S(B|R) - S(AB|R) \tag{8}$$

---

[2]A bosonic system with $N$ modes is classically described by a phase space $V \simeq \mathbb{R}^{2N}$ equipped with an antisymmetric, nondegenerate bilinear form $\Omega : V^* \times V^* \to \mathbb{R}$, known as symplectic form. A linear transformation $M : V \to V$ is called symplectic when it preserves $\Omega$, *i.e.*, $M\Omega M^\intercal = \Omega$.

[3]The minimization in (7) should include also auxiliary quantum systems $R$ with infinite dimension. However, [31, Lemma 7] proves that we can restrict to finite-dimensional $R$ whenever $I(A;B)(\rho)$ is finite.

is the quantum conditional mutual information, and

$$S(A|R) = S(AR) - S(R) \tag{9}$$

is the conditional von Neumann entropy. The squashed entanglement is a faithful entanglement measure, *i.e.*, it is zero iff the state is separable[4], and it does not increase under any composition of local operations performed on the subsystems $A$ and $B$ with the possible help of unlimited classical communication between $A$ and $B$. Moreover, the squashed entanglement of any pure state coincides with the entanglement entropy. The squashed entanglement is one of the two main entanglement measures in quantum communication theory: it provides one of the best known upper bound to the length of a shared secret key that can be generated by two parties holding many copies of the quantum state [28, 32–34] and has applications in recoverability theory [35, 36] and multiparty information theory [37–39].

Let us emphasize that while our rigorous results describe the long-time asymptotics $t \to \infty$ of quadratic Hamiltonians, we will discuss their physical significance and applications in the context of interacting and periodically driven systems, where this asymptotics describes an intermediate phase before the entanglement entropy eventually saturates.

This manuscript is structured as follows: In section 2, we first review the results of [18] and [21] in order to prove the required propositions for our main results, *i.e.*, the linear growth of the entanglement entropy for arbitrary pure initial states, and its extension to squashed entanglement of mixed initial states. In section 3, we use a simple toy model to demonstrate that the inequality (3) will not suffice to prove that the logarithmic growth for meta-stable Hamiltonians found in [19] also applies to arbitrary non-Gaussian states. In section 4, we discuss physical applications and in which sense our long-time asymptotics actually describes an intermediate phase before the entanglement entropy saturates. Finally, we summarize our findings and provide an outlook in section 5.

## 2 Linear growth

In this section, we prove the main result of this manuscript, after we review its two main ingredients: the linear growth for Gaussian initial state proven in [18] and the recently discovered generalized strong subadditivity of the von Neumann entropy proven in [21].

### 2.1 Bosonic quantum systems

We consider a bosonic quantum system $A$ with $N$ modes and classical phase space $V \simeq \mathbb{R}^{2N}$ equipped with an anti-symmetric, non-degerate bilinear form $\Omega : V^* \times V^* \to \mathbb{R}$ defined on the dual phase space. Classical linear observables are elements of the dual phase space $w_1, w_2 \in V^*$ with canonical Poisson brackets $\{w_1, w_2\} = \Omega(w_1, w_2)$. Under quantization, these observables are promoted to operators[5] $\hat{w}_1$ and $\hat{w}_2$ with canonical commutation relations given by $[\hat{w}_1, \hat{w}_2] = i\Omega(w_1, w_2)$.

We can choose a so-called *Darboux* basis of $N$ canonically conjugate pairs $(\hat{q}_i, \hat{p}_i)$ of *quadrature operators*

$$\hat{\xi}^a \equiv (\hat{q}_1, \hat{p}_1, \ldots, \hat{q}_N, \hat{p}_N), \tag{10}$$

such that the following canonical commutation relations are satisfied:

$$\left[\hat{\xi}^a, \hat{\xi}^b\right] = i\Omega_{2N}^{ab} \quad \text{with} \quad \Omega_{2N} \equiv \bigoplus_{i=1}^{N} \begin{pmatrix} 0 & 1 \\ -1 & 0 \end{pmatrix}. \tag{11}$$

---

[4]This is guaranteed when $S(AB)(\rho)$, $S(A)(\rho)$ and $S(B)(\rho)$ are not all infinite [31, Proposition 8].

[5]We will use hats on observables, such as $\hat{w}$, $\hat{\xi}^a$ and $\hat{H}$, but not on density operators $\rho$ and unitaries $U$.

Table 1: *Conventions and notation.* We list the most common symbols and how they are used in this manuscript.

| Symbol | Meaning |
|---|---|
| $V$ | classical phase space |
| $V^*$ | dual phase space (linear observables) |
| $\Omega$ | symplectic form on $V^*$ |
| $\Omega_{2N}$ | $2N$-by-$2N$ matrix representation of $\Omega$ |
| $\hat{\xi}^a$ | quadrature basis $\hat{\xi}^a = (\hat{q}_1, \hat{p}_1, \ldots, \hat{q}_N, \hat{p}_N)$ of $V^*$ |
| $M$ | symplectic transformation $M : V \to V$, such that $M\Omega M^\intercal = \Omega$ |
| $U_M$ | unitary transformation implementing $M$ |
| $\mathcal{U}_M$ | quantum channel $\mathcal{U}_M$, see (5) |
| $\rho$ | general density operator $\rho : \mathcal{H} \to \mathcal{H}$ |
| $\sigma$ | Gaussian density operator |
| $\sigma_{G,z}$ | Gaussian density operator with covariance matrix $G$ and displacement $z$ |
| $z^a$ | displacement vector $z \in V$ of Gaussian state $\sigma_{G,z}$ |
| $G^{ab}$ | covariance matrix of Gaussian state $\sigma_{G,z}$ |
| $J^a{}_b$ | complex structure $J = G\Omega^{-1}$ of Gaussian state $\sigma_{G,z}$ |
| tr, Tr | trace tr on classical phasespace and trace Tr on Hilbert space |
| $S(A)(t)$ | entanglement entropy of subsystem $A$ at time $t$ (of pure state evolved with quadratic Hamiltonian) |
| $I(A;B)(\rho)$ | mutual information of state $\rho$ with respect to subsystems $A$ and $B$ |
| $S_2(A)(\rho)$ | Rényi entropy (of order 2) of state $\rho$ and subsystem $A$ |
| $E_{\mathrm{sq}}(\rho)$ | squashed entanglement of state $\rho$, see (7) |
| $\hat{H}(t)$ | time-dependent Hamiltonian at time $t$ |
| $L(M)$ | limiting matrix of time-dependent symplectic transformation $M$, see (23) |
| $\lambda_\ell$ | Lyapunov exponent for time-dependent $M$ and dual vector $\ell \in V^*$, see (26) |
| $\Lambda_A$ | subsystem exponent for subsystem $A$, see (27) |
| $S_{\mathrm{as}}(C)(\alpha)$ | asymptotic von Neumann entropy of system $C$, see (17) |

A (potentially mixed) Gaussian state $\sigma_{G,z}$ is fully characterized by its covariance matrix $G$ and displacement vector $z$ given by

$$z^a = \mathrm{Tr}\left[\sigma_{G,z}\,\hat{\xi}^a\right] \quad \text{and} \quad G^{ab} = \mathrm{Tr}\left[\hat{\xi}^a\,\sigma_{G,z}\,\hat{\xi}^b + \hat{\xi}^b\,\sigma_{G,z}\,\hat{\xi}^a\right] - 2z^a z^b\,, \tag{12}$$

where $z$ can be an arbitrary phase-space vector $z \in V$, while $G$ is a positive-definite symmetric bilinear form such that $J = G\Omega^{-1}$ has the property that all the eigenvalues of $-J^2$ are larger than one. In particular, the Gaussian state $\sigma_{G,z}$ is a pure state if and only if $J^2 = -\mathbb{1}$, in which case $(\Omega, G, J)$ form a so-called Kähler structure [40–42]. For the sake of a simpler notation, we omit the displacement vector when it is zero, *i.e.*, we define $\sigma_G = \sigma_{G,0}$

All the quantum states with finite average energy[6], which include all the quantum states that can be generated in physical experiments, have a well-defined covariance matrix. However, one could formally construct states for which certain entries of $G$ diverge[7].

The von Neumann entropy and the Rényi entropy of order 2 of a Gaussian state take par-

---

[6]One requires that the expectation value $\langle\psi|\hat{H}|\psi\rangle$ is finite for a quadratic Hamiltonian $\hat{H} = \frac{1}{2}\sum_{a,b}h_{ab}\hat{\xi}^a\hat{\xi}^b$ with symmetric, positive definite bilinear form $h_{ab} > 0$. If this condition is satisfied for one choice of $h_{ab}$, it is satisfied for all and equivalent to requiring that all entries of the covariance matrix $G^{ab}$ are finite.

[7]A simple example is the pure state $|\psi\rangle = \frac{\sqrt{6}}{\pi}\sum_{n=1}^{\infty}\frac{1}{n}|n\rangle$ of a single bosonic degree of freedom written in the number operator basis of a harmonic oscillator, for which the harmonic oscillator energy of $\hat{H} = \frac{E_0}{2}(\hat{q}^2 + \hat{p}^2) = E_0(\hat{n} + \frac{1}{2})$ diverges due to $\langle\psi|\hat{H}|\psi\rangle = \frac{6E_0}{\pi^2}\sum_{n=1}^{\infty}\frac{2n+1}{2n^2} = \infty$.

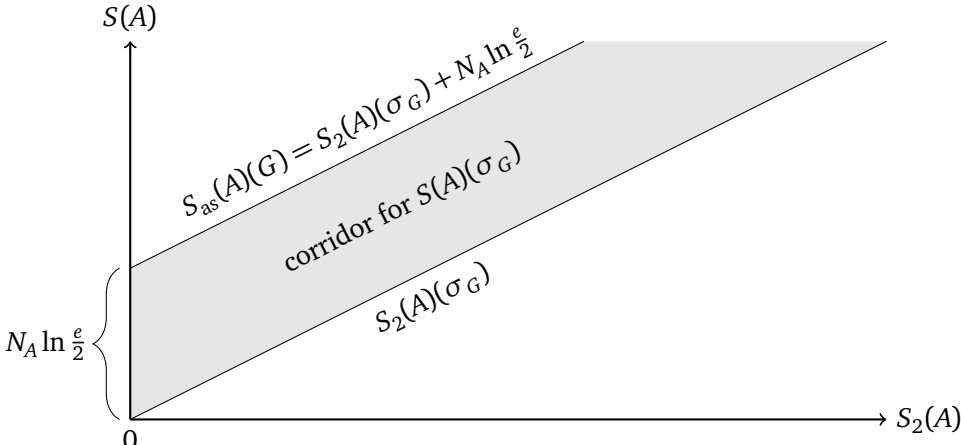

Figure 1: *Bounding corridor for von Neumann entropy of Gaussian states.* We show how the von Neumann entropy $S(A)(\rho)$ of a Gaussian state $\sigma_G$ in a system $A$ is bounded from below by the Rényi entropy $S_2(A)$ and from above by the asymptotic entropy $S_{as}(A)(G)$.

ticularly simple forms when written in terms of $J$, namely

$$S(A)(\sigma_{G,z}) = \operatorname{tr}\left[\frac{\mathbb{1} + \mathrm{i}J}{2}\ln\left|\frac{\mathbb{1} + \mathrm{i}J}{2}\right|\right],$$

$$S_2(A)(\sigma_{G,z}) = \frac{1}{2}\ln\det(\mathrm{i}J), \tag{13}$$

where the Rényi entropy of order 2 of a generic state $\rho$ of $A$ is

$$S_2(A)(\rho) = -\ln\operatorname{Tr}\rho^2, \tag{14}$$

and provides a lower bound to the von Neumann entropy, *i.e.*, for any state $\rho$ of $A$,

$$S_2(A)(\rho) \leq S(A)(\rho). \tag{15}$$

For Gaussian states, the difference between the von Neumann entropy and the Rényi entropy of order 2 is upper bounded by the number of modes:

**Proposition 1.** *The von Neumann entropy and the Rényi entropy of order 2 of any mixed Gaussian state $\sigma$ of the bosonic quantum system $A$ with $N$ modes satisfy the bounds*

$$S_2(A)(\sigma) \leq S(A)(\sigma) \leq S_2(A)(\sigma) + N\ln\frac{e}{2}. \tag{16}$$

*Proof.* This result is well-known and appeared in various forms in the literature. Among other places, it is shown in [18, Section 6.1]. □

Proposition 1 allows to replace the von Neumann entropy by the Rényi entropy of order 2 by only making a finite error independent of the quantum state (though increasing for larger systems). We therefore define the *asymptotic von Neumann entropy*

$$S_{as}(A)(G) = S_2(A)(\sigma_G) + N_A\ln\frac{e}{2}, \tag{17}$$

which gives for each covariance matrix $G$ the corresponding upper bound from Proposition 1. This bound becomes exact when all symplectic eigenvalues of $G$ are large [21, Lemma 9].

Using the previous result, we will often replace the von Neumann entropy of a Gaussian state by its Rényi entropy of order 2, whose asymptotic growth can be more easily analyzed by the following geometric interpretation:

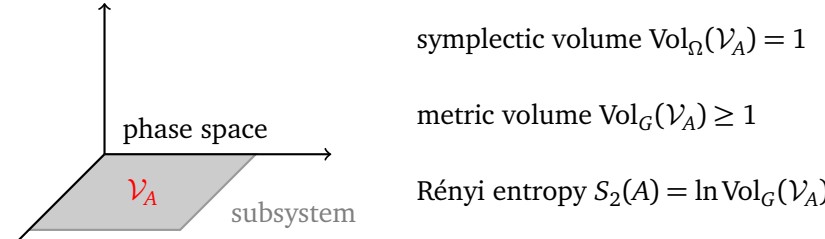

**Figure 2:** *Geometric interpretation of Rényi entropy.* We can interpret the Rényi entropy (of order 2) as the logarithm of the metric volume of a parallelepiped on the subspace $A^* \subset V^*$, whose symplectic volume is equal to 1. The metric volume is calculate by restrcting the covariance matrix $G$ of the given state to the subspace $A$ that contains $\mathcal{V}_A$.

**Proposition 2.** *The Rényi entropy of order 2 of the Gaussian state $\sigma_{G,z}$ of A is equal to the logarithm of the metric volume with respect to G of any region $\mathcal{V} \subset V^*$ with unit symplectic volume, i.e.,*

$$S_2(A)(\sigma) = \ln \mathrm{Vol}_G(\mathcal{V}). \tag{18}$$

*Proof.* The simple proof can be found in [18, Section 6.2] and gives the Rényi entropy of order 2 a concrete geometric interpretation. We recall that the Rényi entropy of order 2 can be written as determinant [17]

$$S_2(\sigma) = \frac{1}{2} \ln \det |iJ|, \tag{19}$$

where $J = -G\Omega^{-1}$. We can always choose a basis $(v^1, \ldots, v^{2N})$ of $V^*$ where $\Omega$ has the standard form (11), such that $\det \Omega = 1$. Note that this implies that the parallelepiped spanned by the chosen basis vectors has unit volume with respect to the symplectic volume form. The matrix entries $G_{ij} = G(v_i, v_j)$ can be understood as the inner products $\langle v_i, v_j \rangle_G$ defined by $G$, which is also known as Gram matrix. It is well-known that the determinant of a Gram matrix

$$\det G = \det \begin{pmatrix} \langle v_1, v_1 \rangle_G & \cdots & \langle v_1, v_{2N} \rangle_G \\ \vdots & \ddots & \vdots \\ \langle v_{2N}, v_1 \rangle_G & \cdots & \langle v_{2N}, v_{2N} \rangle_G \end{pmatrix} = (\mathrm{Vol}_G(\mathcal{V}))^2 \tag{20}$$

equals the square of the geometric volume of the respective parallelepiped (spanned by the chosen basis, measured with respect to the inner product $G$). Therefore, the prefactor of $\frac{1}{2}$ will cancel the square and we arrive at (18). □

After this review of Gaussian states and their entropies, we will now turn to the dynamics under quadratic Hamiltonians. Given such a Hamiltonian

$$\hat{H}(t) = \frac{1}{2} \sum_{a,b=1}^{N} h_{ab}(t) \hat{\xi}^a \hat{\xi}^b + \sum_{a=1}^{N} f_a(t) \hat{\xi}^b, \tag{21}$$

we define the symplectic generator $K(t) = \Omega h(t)$ that gives rise to the time-dependent symplectic group element $M(t)$ written as time-ordered exponential

$$M(t) = \mathcal{T} \exp \int_0^t K(t') \, dt'. \tag{22}$$

For every time-dependent symplectic transformation $M(t)$, we define the limiting matrix[8]

$$L(M) = \lim_{t \to \infty} \frac{\ln\left(M(t)M(t)^{\mathsf{T}}\right)}{2t}, \tag{23}$$

provided this limit exists. The eigenvectors $\ell$ of $L(M)$ are elements of the dual phase space $V^*$ and we can always choose an orthonormal eigenbasis

$$\mathcal{D} = (\ell^1, \dots, \ell^{2N}), \tag{24}$$

whose elements we sort such that the associated eigenvalues $\lambda_i$ satisfy $\lambda_1 \geq \cdots \geq \lambda_{2N}$. The eigenvalues $\lambda_i$ are also called *Lyapunov exponents* of the respective basis vector $\ell^i$ (see [18, Appendix A.2]).

The basis $\mathcal{D}$ gives rise to the sequence of subspaces

$$V_{2N}^* \subset V_{2N-1}^* \subset \cdots \subset V_2^* \subset V_1^* = V^*, \tag{25}$$

with $V_k^* = \mathrm{span}(\ell^1, \dots \ell^k)$, which characterizes the long-time behavior of elements on. For any $\ell \in V^*$, we define its Lyapunov exponents as

$$\lambda_\ell = \lim_{t \to \infty} \ln\|M(t)^{\mathsf{T}}\ell\| = \lambda_k \quad \text{for} \quad \ell \in V_k^* \setminus V_{k+1}^*. \tag{26}$$

There is the notion of a *regular* Hamiltonian system (see [18, Appendix A.3]), which is characterized by the property that the symplectic flow $M(t)$ has well-defined Lyapunov exponents that appear in conjugate pairs, such that $\lambda_k = -\lambda_{2N+1-k}$. This is a rather natural assumption, as most examples violating these assumptions are rather unphysical (*e.g.*, due to above-exponential growth) and in particular, any time-independent quadratic Hamiltonian is *regular*.

A crucial question will be how the metric volume (with respect to a positive-definite, bilinar form $G$) of a parallelepiped $\mathcal{V} \subset V^*$ behaves when we evolve it with the symplectic transformation $M(t)$, *i.e.*, what is $\ln \mathrm{Vol}_G(M(t)^{\mathsf{T}}\mathcal{V})$? As it turns out, its leading order depends only on the subspace of $V^*$ spanned by $\mathcal{V}$, and is independent of $G$ and of the shape of $\mathcal{V}$.

## 2.2 Entanglement growth for Gaussian states

Let $A$ and $B$ be bosonic quantum systems with $N_A$, $N_B$ modes, phase spaces $V_A$, $V_B$, symplectic forms $\Omega_A$, $\Omega_B$ and Hilbert spaces $\mathcal{H}_A$, $\mathcal{H}_B$, respectively. Their union is the bipartite bosonic quantum system $AB$ with $N = N_A + N_B$ modes, phase space $V = V_A \oplus V_B$, symplectic form $\Omega = \Omega_A \oplus \Omega_B$ and Hilbert space $\mathcal{H} = \mathcal{H}_A \otimes \mathcal{H}_B$. We say that $A$ and $B$ are subsystems of $AB$. Given a Gaussian state $\sigma_{G,z}$ of $AB$, its marginal state on $A$ is the Gaussian state $\sigma_{G_A, z_A}$, where $G_A$ is the restriction of $G$ to $V_A^*$ and $z_A$ is the projection of $z$ onto $V_A$.

For a given time-dependent symplectic transformation $M(t)$, we associate to the subsystem $A$ the exponent

$$\Lambda_A = \lim_{t \to \infty} \frac{\ln \mathrm{Vol}_G(M(t)^{\mathsf{T}}\mathcal{V})}{t}, \tag{27}$$

where $\mathcal{V} \subset V_A^*$ is any parallelepiped whose span is equal to $V_A^*$ and $G$ is any positive-definite bilinear form on $V^*$. The subsystem exponent does not depend on $G$ and can be computed explicitly from the Lyapunov basis $\mathcal{D}$ and the Lyapunov exponents $\lambda_k$, as follows.

---

[8]At this point, we perform all computations in a fixed basis, so that we can represent $M(t)$ as matrix. Otherwise, the limiting matrix will explicitly depend on an auxiliary inner product $G$, which we chose to be the identity in our basis.

$$\Lambda_A = \lim_{t\to\infty} \log \mathrm{Vol}_G(M(t)^\mathsf{T}\mathcal{V}) = \sum_{k=1}^{2N_B} \lambda_{i_k}$$

Figure 3: *Subsystem exponent due to phase space stretching.* The subsystem exponent (27) describes the exponential volume growth of an initial parallelepiped $\mathcal{V}$ whose span is equal to $V_B^*$ under the symplectic time evolution $M(t)^\mathsf{T}$. This figure is based on the respective figure in [18].

**Proposition 3.** *Given a bipartite bosonic quantum system AB and a regular Hamiltonian system characterized by $M(t)$ with Lyapunov spectrum $(\lambda_1, \ldots, \lambda_{2N})$ and Lyapunov basis $\mathcal{D} = (\ell^1, \ldots, \ell^{2N})$, the subsystem exponent $\Lambda_A$ of A can be computed as follows:*

1. *Choose a Darboux basis $\mathcal{D}_A = (\theta^1, \ldots, \theta^{2N_A})$ of $V_A^*$, i.e., the restricted symplectic form $\Omega_A$ takes the form of (11).*

2. *Compute the unique transformation matrix $F$ that expresses $\mathcal{D}_A$ in terms of the Lyapunov basis $\mathcal{D} = (\ell^1, \ldots, \ell^{2N})$:*

$$
\begin{pmatrix} \theta^1 \\ \vdots \\ \theta^{2N_A} \end{pmatrix} = \begin{pmatrix} \underbrace{\boxed{\begin{matrix} F_1^1 \\ \vdots \\ F_1^{2N_A} \end{matrix}}}_{\vec{F}_1} & \cdots & \underbrace{\boxed{\begin{matrix} F_{2N}^1 \\ \vdots \\ F_{2N}^{2N_A} \end{matrix}}}_{\vec{F}_{2N}} \end{pmatrix} \begin{pmatrix} \ell^1 \\ \vdots \\ \ell^{2N} \end{pmatrix} \quad . \tag{28}
$$

   *We refer to the 2N columns of F as $\vec{F}_i$.*

3. *Find the first $2N_A$ linearly independent [9] columns $\vec{t}_i$ of T which we can label by $\vec{t}_{i_k}$ with k ranging from 1 to $2N_A$. The result is a map $k \mapsto i_k \in (1, \ldots, 2N)$ with $i_{k+1} > i_k$.*

*The subsystem exponent is then given by the sum $\Lambda_A = \sum_{k=1}^{2N_A} \lambda_{i_k}$ over the $2N_A$ Lyapunov exponents $\lambda_{i_k}$, where the index $i_k$ is defined above.*

*Remark.* For almost all subsystems (except a measure zero set), the subsystem exponent is given by the sum of the largest $2N_A$ Lyapunov exponents, *i.e.*, $\Lambda_A = \sum_{i=1}^{2N_A} \lambda_i$.

*Proof.* This result was proven in the context of entanglement growth as [18, Theorem 3 (Subsystem exponent)]. The key idea is that $M(t)$ will stretch any $2N_A$-dimensional volume region (including any parallelepiped) dominantly into those $2N_A$ directions with the largest Lyapunov exponents, provided that there are directions in the subspace $V_A^*$ that are linearly dependent

---

[9]Here, we mean that $\vec{F}_i$ cannot be expressed as a linear combination of the vectors $(\vec{F}_1, \ldots, \vec{F}_{i-1})$ standing to the left in the matrix $F$.

on those stretching directions. Unsurprisingly, the generic situation (*i.e.*, applicable to all subsystems except for a subset of measure zero) the subspace $V_A^*$ will have overlap with all stretching directions, so that the $2N_A$ largest ones dominate and we get $\Lambda_A = \sum_{i=1}^{2N_A} \lambda_i$, as explained in [18, Theorem 4 (Subsystem exponent – generic subsystem)]. □

The main result of [18] was then derived by combining the reviewed propositions and theorems to understand the large time asymptotics of the entanglement entropy for arbitrary bosonic Gaussian states.

**Proposition 4.** *Given a quadratic time-dependent Hamiltonian $\hat{H}(t)$ and a subsystem $A$ with subsystem exponent $\Lambda_A$, the long-time behavior of the entanglement entropy of the subsystem is*

$$S(A)(t) = \Lambda_A t + o(t), \tag{29}$$

*for all initial Gaussian states. Moreover, this asymptotics also provides an upper bound for non-Gaussian initial states with finite covariance matrix.*

*Proof.* This result follows by combining Proposition 1, Proposition 2 and Proposition 3 and was proven in [18, Theorem 1]. The proof follows three steps:

1. Proposition 1 states that the entanglement entropy $S(A)(t)$ is bounded according to $S_2(A)(t) \leq S(A)(t) \leq S_2(A)(t) + N_A \ln \frac{e}{2}$. Provided that we can show that $S_2(A)(t) = \Lambda_A t + o(t)$ as $t \to \infty$, the same asymptotics will also apply to $S(A)(t)$.

2. Proposition 2 provides a geometric interpretation of the Rényi entropy of order 2 in terms of the geometric volume of a parallelepiped $\mathcal{V}$ with unit symplectic volume. The covariance matrix of the initial state will evolve with time as $G(t) = M(t)G(0)M(t)^{\mathsf{T}}$. This allows us to rewrite the Rényi entropy of order 2 as

$$S_2(A)(t) = \log \mathrm{Vol}_{G(t)}(\mathcal{V}) = \log \mathrm{Vol}_{G(0)}(M(t)^{\mathsf{T}}\mathcal{V}), \tag{30}$$

where we notice that it is the same to measure the volume of the initial parallelepiped $\mathcal{V}$ with respect to the time-dependent inner product $G(t)$ or to measure the volume of the time-dependent parallelepiped $\mathcal{V}(t) = M(t)^{\mathsf{T}}\mathcal{V}$ with respect to the initial inner product $G(0)$. We then recognize the resulting growth rate as the subsystem exponent $\Lambda_A$ from (27).

3. Proposition 3 then shows how this subsystem exponent is calculated in practice and that it actually turns out to be independent of the initial covariance matrix $G(0)$.

This result already provides an upper bound on the entanglement growth for *arbitrary* pure initial states with finite covariance matrix (see also [18, Section 2.3]). This is due to the fact that the time evolution under a quadratic Hamiltonian changes the covariance matrix according to $G(t) = M(t)G(0)M(t)^{\mathsf{T}}$, *i.e.*, just as if the state were Gaussian. Moreover, it is well-known that among all the mixed states of the subsystem $A$ with covariance matrix $[G(t)]_A$, the Gaussian state has the maximal von Neumann entropy. As our proof shows that the associated entropy of Gaussian states grows at most with the rate $\Lambda_A$, this serves as an upper bound on the entanglement growth for all states. □

In summary, following [18] we showed that the entanglement entropy grows linearly with rate $\Lambda_A$, which also serves as an upper bound for any non-Gaussian state with finite covariance matrix.

## 2.3 Generalized strong subadditivity

The main ingredient of the lower bound for the time scaling of the entanglement entropy is the generalized strong subadditivity of the von Neumann entropy for bosonic quantum systems proved in [21]. Let $C$ be a bosonic quantum system with $N$ modes. For each $i = 1, \dots, k$, let $F_i \in \mathbb{R}^{2N_i \times 2N}$ be a linear map[10] that preserves the symplectic form, *i.e.*, such that

$$F_i \, \Omega_{2N} \, F_i^{\mathsf{T}} = \Omega_{2N_i} \,. \tag{31}$$

Each $F_i$ identifies the subsystem $C_i$ of $C$ with $N_i$ modes with dual phase space $V_{C_i}^* = \operatorname{Im} F_i^{\mathsf{T}}$. $C_i$ is associated to the quadratures

$$\left\{ \sum_{b=1}^{2N_i} (F_i)^a{}_b \, \hat{\xi}^b : a = 1, \dots, 2N_i \right\} \,, \tag{32}$$

which thanks to the condition (31) satisfy the canonical commutation relations. Let $\mathbf{p} \in \mathbb{R}^k_{\geq 0}$ satisfy the scaling condition

$$N = \sum_{i=1}^{k} p_i \, N_i \,. \tag{33}$$

Ref. [21] considers the following maximization problem:

$$f(\mathbf{p}) = \sup_{\rho} \left\{ S(C)(\rho) - \sum_{i=1}^{k} p_i \, S(C_i)(\rho) \right\} \,, \tag{34}$$

where the supremum is performed over all the states $\rho$ of $C$ with finite covariance matrix, and proves that such infinitely dimensional optimization can be reduced to the following finite-dimensional optimization over $2N \times 2N$ positive definite matrices:

$$f(\mathbf{p}) = \sup_{G \in \mathbb{R}^{2N \times 2N}_{>0}} \left( S_{\mathrm{as}}(C)(G) - \sum_{i=1}^{k} p_i \, S_{\mathrm{as}}(C_i)(G) \right) \,, \tag{35}$$

where the asymptotic von Neumann entropy of $C$ and of each subsystem $C_i$ associated to $G \in \mathbb{R}^{2N \times 2N}_{>0}$ are

$$S_{\mathrm{as}}(C)(G) = \frac{1}{2} \ln \det \frac{e\,G}{2} \,, \qquad S_{\mathrm{as}}(C_i)(G) = \frac{1}{2} \ln \det \left( \frac{e}{2} F_i \, G \, F_i^{\mathsf{T}} \right) \,. \tag{36}$$

*Remark.* If $G$ is a valid covariance matrix of a quantum state, the asymptotic von Neumann entropy of $G$ is equal to the Rényi entropy of order 2 of $\sigma_G$ up to a constant:

$$S_{\mathrm{as}}(C)(G) = S_2(C)(\sigma_G) + N \ln \frac{e}{2} \,. \tag{37}$$

The main idea of the proof of (35) is perturbing the state with the quantum heat semigroup. The same idea has been crucial in the proofs of several quantum versions of the Entropy Power Inequality [43–54], of which (35) can be considered a generalization. Let $\bar{G}$ achieve the maximum in (35) (if the maximum is not achieved, the result can be obtained with a limiting argument). Ref. [21] considers a generic state $\rho$ of $C$ with finite covariance matrix, and evolves it with the time evolution induced by the quantum heat semigroup that adds classical Gaussian noise with covariance matrix proportional to $\bar{G}$. For any $t \geq 0$, let $\rho_t$ be the state at

---

[10]For the sake of a simpler notation, in the reminder of this section we always choose a basis such that the symplectic form has the canonical form (11) and consider all linear and bilinear forms as matrices in such basis.

time $t$. On the one hand, Ref. [21] proves that the quantity to be maximized increases with time:

$$\frac{d}{dt}\left(S(C)(\rho_t)-\sum_{i=1}^{k}p_i\,S(C_i)(\rho_t)\right)\geq 0\,,\tag{38}$$

such that

$$S(C)(\rho)-\sum_{i=1}^{k}p_i\,S(C_i)(\rho)\leq\lim_{t\to\infty}\left(S(C)(\rho_t)-\sum_{i=1}^{k}p_i\,S(C_i)(\rho_t)\right).\tag{39}$$

On the other hand, Ref. [21] proves that in the limit of infinite time, the maximum postulated in (35) is always achieved, *i.e.*, for any initial state $\rho$,

$$\lim_{t\to\infty}\left(S(C)(\rho_t)-\sum_{i=1}^{k}p_i\,S(C_i)(\rho_t)\right)=S_{\mathrm{as}}(C)(\bar{G})-\sum_{i=1}^{k}p_i\,S_{\mathrm{as}}(C_i)(\bar{G})\,,\tag{40}$$

and the claim follows. In particular, the proof can be applied when $\rho$ is Gaussian. Since the quantum heat semigroup preserves the set of the Gaussian states, the supremum in (34) can always be achieved by a sequence of Gaussian states, and can therefore be restricted to Gaussian states:

$$f(\mathbf{p})=\sup_{\sigma\,\mathrm{Gaussian}}\left\{S(C)(\sigma)-\sum_{i=1}^{k}p_i\,S(C_i)(\sigma)\right\}.\tag{41}$$

Let us now provide an intuition of where the expression (35) comes from. The quantum heat semigroup described above acts on a Gaussian state $\sigma$ by adding $t\,\bar{G}$ to the covariance matrix. If $G$ is a valid covariance matrix for a quantum state, $S_{\mathrm{as}}(G)$ approximates the entropy of the Gaussian state $\sigma_G$ when all ther symplectic eigenvlaues of $G$ are large. Indeed, if $\nu_{\mathrm{min}}^2\geq 1$ is the minumum eigenvalue of $-\left(G\,\Omega^{-1}\right)^2$, we have [21, Lemma 9]

$$S_{\mathrm{as}}(C)(G)-\frac{N}{\nu_{\mathrm{min}}^2}\ln\frac{e}{2}\leq S(C)(\sigma_G)\leq S_{\mathrm{as}}(C)(G)\,.\tag{42}$$

Since $\bar{G}$ is positive definite, in the limit of infinite time the covariance matrix of $\sigma_t$ can be approximated with $t\,\bar{G}$, and the entropy of $\sigma_t$ is approximately

$$S(C)(\sigma_t)\simeq S_{\mathrm{as}}(t\,\bar{G})=S_{\mathrm{as}}(\bar{G})+N\ln t\,.\tag{43}$$

Similarly, for any $i=1,\ldots,k$, the covariance matrix of the marginal of $\sigma_t$ over $C_i$ can be approximated with $t\,F_i\,\bar{G}\,F_i^{\mathsf{T}}$, and

$$S(C_i)(\sigma_t)\simeq S_{\mathrm{as}}(C_i)(t\,\bar{G})=S_{\mathrm{as}}(C_i)(\bar{G})+N_i\ln t\,.\tag{44}$$

Thanks to the scaling condition (33), the terms proportional to $\ln t$ cancel each other and we get as wanted

$$S(C)(\sigma_t)-\sum_{i=1}^{k}p_i\,S(C_i)(\sigma_t)\simeq S_{\mathrm{as}}(C)(\bar{G})-\sum_{i=1}^{k}p_i\,S_{\mathrm{as}}(C_i)(\bar{G})\,.\tag{45}$$

In this paper, we consider the setup with $k=4$ subsystems. Let $N=N_A+N_B$, let

$$F_A=\left(\begin{array}{cc}\mathbb{1}_{2N_A} & 0_{2N_A\times 2N_B}\end{array}\right)\tag{46}$$

select the first $2N_A$ components, and let

$$F_B=\left(\begin{array}{cc}0_{2N_B\times 2N_A} & \mathbb{1}_{2N_B}\end{array}\right)\tag{47}$$

select the last $2N_B$ components, such that $A$ and $B$ are complementary subsystems of $C$, *i.e.*, $C = AB$. Let $M \in \mathrm{Sp}(2N, \mathbb{R})$ be a symplectic matrix and let $F_{A'} = F_A M$ and $F_{B'} = F_B M$, such that the subsystems $A'$ and $B'$ correspond to the subsystems $A$ and $B$ after the application of the transformation $M$. Then, setting

$$p_A = p_B = p_{A'} = p_{B'} = \frac{1}{2} \tag{48}$$

in (35) and (41), we get

**Proposition 5.** *Let AB be a bipartite bosonic quantum system with $N = N_A + N_B$ modes, where $A$ and $B$ have $N_A$ and $N_B$ modes, respectively. Then, for any symplectic transformation $M \in \mathrm{Sp}(2N, \mathbb{R})$, the mutual information $I(A;B)$ of any quantum state $\rho$ of AB with finite covariance matrix satisfies*

$$
\begin{aligned}
I(A;B)(\rho) + I(A;B)(\mathcal{U}_M(\rho)) &\geq \inf_{\sigma \text{ Gaussian}} \left( I(A;B)(\sigma) + I(A;B)(\mathcal{U}_M(\sigma)) \right) \\
&= \inf_{G \in \mathbb{R}_{>0}^{2N \times 2N}} \left( I_{\mathrm{as}}(A;B)(G) + I_{\mathrm{as}}(A;B)(M\,G\,M^\intercal) \right),
\end{aligned}
\tag{49}
$$

*where we have defined for any $G \in \mathbb{R}_{>0}^{2N \times 2N}$*

$$I_{\mathrm{as}}(A;B)(G) = S_{\mathrm{as}}(A)(G) + S_{\mathrm{as}}(B)(G) - S_{\mathrm{as}}(AB)(G). \tag{50}$$

The supremum in (41) is always achieved in the limit of infinite covariance matrix, and is therefore never achieved by a Gaussian state with finite covariance matrix. On the contrary, Ref. [21] proves that the supremum in (35) is achieved iff there exists $\bar{G} \in \mathbb{R}_{>0}^{2N \times 2N}$ satisfying

$$\bar{G}^{-1} = \sum_{i=1}^{k} p_i F_i^\intercal \left( F_i\, \bar{G}\, F_i^\intercal \right)^{-1} F_i. \tag{51}$$

Moreover, if such $\bar{G}$ exists, the supremum in (35) is achieved by $G = \bar{G}$. In general, an analytical solution of (51) cannot be found. However, in the setup of Proposition 5, if the symplectic matrix $M$ is also positive definite, such solution is given by $\bar{G} = M^{-1}$. We then get

**Proposition 6.** *In the setup of Proposition 5, if the symplectic transformation $M$ is also positive definite, the mutual information $I(A;B)$ of any quantum state $\rho$ of AB with finite covariance matrix satisfies*

$$I(A;B)(\rho) + I(A;B)(\mathcal{U}_M(\rho)) \geq 2\, I_{\mathrm{as}}(A;B)(M). \tag{52}$$

*Proof.* From [21, Proposition 19] we get

$$I(A;B)(\rho) + I(A;B)(\mathcal{U}_M(\rho)) \geq I_{\mathrm{as}}(A;B)(M^{-1}) + I_{\mathrm{as}}(A;B)(M). \tag{53}$$

Since $M$ is both symplectic and symmetric, we have

$$M^{-1} = \Omega_{2N}^{-1} M\, \Omega_{2N} = \Omega_{2N}^\intercal M\, \Omega_{2N}, \tag{54}$$

and $\Omega_{2N}$ is symplectic and block-diagonal with respect to the decomposition $\mathbb{R}^{2N} = \mathbb{R}^{2N_A} \oplus \mathbb{R}^{2N_B}$. Therefore, we have

$$
\begin{aligned}
I_{\mathrm{as}}(A;B)(M^{-1}) &= S_{\mathrm{as}}(A)(M^{-1}) + S_{\mathrm{as}}(B)(M^{-1}) - S_{\mathrm{as}}(AB)(M^{-1}) \\
&= S_{\mathrm{as}}(A)(M) + S_{\mathrm{as}}(B)(M) - S_{\mathrm{as}}(AB)(M) = I_{\mathrm{as}}(A;B)(M).
\end{aligned}
\tag{55}
$$

The claim follows. $\qquad\square$

All the results of Ref. [21] are actually proved in the more general setup where all the entropies are conditioned on an arbitrary quantum system $R$ with separable Hilbert space. The generalized version of the maximization problem (34) is

$$f(\mathbf{p}) = \sup_{\rho} \left\{ S(C|R)(\rho) - \sum_{i=1}^{k} p_i S(C_i|R)(\rho) \right\}, \tag{56}$$

where the supremum is performed over all the states $\rho$ of the joint quantum system $CR$ such that $\rho_C$ has finite covariance matrix and $\rho_R$ has finite entropy. Ref. [21] proves that the supremum in (56) coincides with the supremum in (34), and is therefore given by (35). Therefore, Proposition 6 can be generalized as follows:

**Proposition 7.** *Let AB be a bipartite bosonic quantum system with $N = N_A + N_B$ modes, where A and B have $N_A$ and $N_B$ modes, respectively, and let R be an arbitrary quantum system with separable Hilbert space. Let $M \in \mathrm{Sp}(2N, \mathbb{R})$ be a symplectic matrix that is also positive definite. Then, the conditional mutual information $I(A; B|R)$ of any quantum state $\rho$ of ABR such that $\rho_{AB}$ has finite covariance matrix and $\rho_R$ has finite entropy satisfies*

$$I(A; B|R)(\rho) + I(A; B|R)(\mathcal{U}_M(\rho)) \geq 2 I_{\mathrm{as}}(A; B)(M). \tag{57}$$

### 2.4 Entropy growth for pure states

We now consider again a bipartite bosonic quantum system $AB$ with $N = N_A + N_B$ modes and a quadratic Hamiltonian $\hat{H}(t)$ as in (21) giving rise to a symplectic transformation $M(t)$, but this time we only assume that the initial state is pure and has a finte covariance matrix. We decompose $M(t)$ as

$$M(t) = T(t)u(t) \quad \text{with} \quad T(t) = \sqrt{M(t)M(t)^{\intercal}} \quad \text{and} \quad u(t) = M(t)T(t)^{-1}, \tag{58}$$

such that $T(t)$ is both symplectic and positive definite, and $u(t)$ is both symplectic and orthogonal. We will now be able to relate the limiting matrices of $M(t)$ and $T(t)$.

**Proposition 8.** *Given a one-parameter family of symplectic transformations $M(t)$ and its positive symmetric part $T(t)$ as defined in (58), both $M(t)$ and $T(t)$ have the same Lyapunov spectrum and Lyapunov basis, i.e., $L(M) = L(T)$. Moreover, $L\left(\sqrt{T}\right) = L(T)/2$, i.e., $\sqrt{T(t)}$ has the same Lyapunov basis as $M(t)$ and $T(t)$, and its Lyapunov exponents are half of the Lyapunov exponents of $M(t)$ and $T(t)$.*

*Proof.* The first claim follows straightforward from the fact that both Lyapunov spectrum and Lyapunov basis are defined with respect to the dual transformations $M(t)^{\intercal}$ and $T(t)^{\intercal}$ acting on the dual phase space $V^*$. The second claim follows since

$$L\left(\sqrt{T}\right) = \lim_{t \to \infty} \frac{\ln T(t)}{2t} = \frac{L(T)}{2}, \tag{59}$$

where we use that $T(t)$ is positive definite. $\qquad \square$

**Corollary 1.** *Let $\Lambda_A$ be the exponent of the subsystem A with respect to the time evolution induced by $M(t)$. Then, the subsystem exponent of A with respect to $T(t)$ is equal to $\Lambda_A$, and the exponent with respect to $\sqrt{T(t)}$ is equal to $\Lambda_A/2$.*

Combining Proposition 6 with the decomposition (58), we can get a lower bound to the entanglement entropy generated by a generic symplectic transformation:

**Proposition 9.** *Let $T$ be a symplectic and positive definite matrix, let $u$ be a symplectic and orthogonal matrix, and let $M = T u$. Then, for any pure quantum state $\rho$ of AB with finite covariance matrix G we have*

$$S(A)(\mathcal{U}_M(\rho)) \geq S_{\mathrm{as}}(A)(T) + S_{\mathrm{as}}(B)(T) - N \ln \frac{e}{2} - N_A \ln \frac{e \, \|G\|_\infty}{2}. \tag{60}$$

*Proof.* We have

$$
\begin{aligned}
S(A)(\mathcal{U}_M(\rho)) &= S(A)(\mathcal{U}_T(\mathcal{U}_u(\rho))) + S(A)(\mathcal{U}_u(\rho)) - S(A)(\mathcal{U}_u(\rho)) \\
&\overset{(a)}{\geq} \frac{1}{2} I(A;B)(\mathcal{U}_T(\mathcal{U}_u(\rho))) + \frac{1}{2} I(A;B)(\mathcal{U}_u(\rho)) - S(A)(\sigma_{uGu^\intercal}) \\
&\overset{(b)}{\geq} I_{\mathrm{as}}(A;B)(T) - S(A)(\sigma_{\|G\|_\infty \mathbb{1}_{2N}}) \\
&\overset{(c)}{\geq} S_{\mathrm{as}}(A)(T) + S_{\mathrm{as}}(B)(T) - N \ln \frac{e}{2} - S(A)(\sigma_{\|G\|_\infty \mathbb{1}_{2N}}) \\
&\geq S_{\mathrm{as}}(A)(T) + S_{\mathrm{as}}(B)(T) - N \ln \frac{e}{2} - N_A \ln \frac{e \, \|G\|_\infty}{2}.
\end{aligned}
\tag{61}
$$

(a) follows observing that both $\mathcal{U}_T(\mathcal{U}_u(\rho))$ and $\mathcal{U}_u(\rho)$ are pure states and that Gaussian states maximize the entropy among all the states with the same covariance matrix. (b) follows from Proposition 6 and observing that the entropy of a Gaussian state is an increasing function of the covariance matrix. (c) follows since $\det T = 1$. The claim follows. $\qquad\square$

This is leads us to the main result of the present paper for pure states.

**Theorem 1** (Linear growth of entanglement entropy)**.** *For any initial pure state $\rho$ with finite covariance matrix and for any time-dependent quadratic Hamiltonian $\hat{H}(t)$ inducing a symplectic evolution $M(t)$ such that the limiting matrix (23) exists, the entanglement entropy with respect to the bipartition AB grows asymptotically as*

$$S(A)(\rho(t)) = \Lambda_A t + o(t) \quad as \quad t \to \infty, \tag{62}$$

*where the subsystem exponent $\Lambda_A$ is independent of the initial state $\rho$ and can be computed according to Proposition 3.*

*Proof.* On the one hand, Proposition 4 implies

$$S(A)(\rho(t)) \leq \Lambda_A t + o(t). \tag{63}$$

On the other hand, let $\sigma_{\mathbb{1}}$ be the vacuum state of $AB$. We have from Proposition 9 and Corollary 1

$$
\begin{aligned}
S(A)(\rho(t)) &\geq S_{\mathrm{as}}(A)(T(t)) + S_{\mathrm{as}}(B)(T(t)) + O(1) \\
&= S_2(A)\left(\mathcal{U}_{\sqrt{T(t)}}(\sigma_{\mathbb{1}})\right) + S_2(B)\left(\mathcal{U}_{\sqrt{T(t)}}(\sigma_{\mathbb{1}})\right) + O(1) = \Lambda_A t + o(t),
\end{aligned}
\tag{64}
$$

and the claim follows. $\qquad\square$

## 2.5 Squashed entanglement growth for mixed states

Combining Proposition 7 with the decomposition (58), we can get both an upper and a lower bound to the squashed entanglement generated by a generic symplectic transformation:

**Proposition 10.** *Let $T$ be a symplectic and positive definite matrix, let $u$ be a symplectic and orthogonal matrix, and let $M = T u$. Then, for any (generally mixed) quantum state $\rho$ of $A$ with finite covariance matrix $G$ we have*

$$E_{\mathrm{sq}}(\mathcal{U}_M(\rho)) \le \frac{1}{2} S_{\mathrm{as}}(A)(T^2) + \frac{1}{2} S_{\mathrm{as}}(B)(T^2) + \frac{N}{2} \ln \|G\|_\infty \,,$$
$$E_{\mathrm{sq}}(\mathcal{U}_M(\rho)) \ge S_{\mathrm{as}}(A)(T) + S_{\mathrm{as}}(B)(T) - 2N \ln \frac{e}{2} - N \ln \|G\|_\infty \,. \tag{65}$$

*Proof.* The proof proceeds along the same lines as the proof of Proposition 9.

*Upper bound:* We have from the subadditivity of the entropy

$$E_{\mathrm{sq}}(\mathcal{U}_M(\rho)) \le \frac{1}{2} I(A;B)(\mathcal{U}_M(\rho)) \le \frac{S(A)(\mathcal{U}_M(\rho)) + S(B)(\mathcal{U}_M(\rho))}{2} \,. \tag{66}$$

Since Gaussian states maximize the entropy among all the states with the same covariance matrix, we have

$$S(A)(\mathcal{U}_M(\rho)) \le S(A)(\sigma_{MGM^{\mathsf{T}}}) \le S_{\mathrm{as}}(A)(M\,G\,M^{\mathsf{T}}) \le S_{\mathrm{as}}(A)(\|G\|_\infty\,M\,M^{\mathsf{T}})$$
$$= S_{\mathrm{as}}(A)(T^2) + N_A \ln \|G\|_\infty \,, \tag{67}$$

and the claim follows.

*Lower bound:* Let $R$ be an arbitrary finite-dimensional quantum system, and let $\tilde{\rho}$ be a quantum state of the joint quantum system $AR$ such that $\mathrm{Tr}_R \tilde{\rho} = \rho$. We notice that $\mathcal{U}_M(\tilde{\rho})$ is an extension of $\mathcal{U}_M(\rho)$, i.e., $\mathrm{Tr}_R \mathcal{U}_M(\tilde{\rho}) = \mathcal{U}_M(\rho)$. We have from the subadditivity of the entropy

$$I(A;B|R)(\mathcal{U}_u(\tilde{\rho})) = S(A|R)(\mathcal{U}_u(\tilde{\rho})) - S(A|BR)(\mathcal{U}_u(\tilde{\rho})) \le 2\,S(A)(\mathcal{U}_u(\tilde{\rho}))$$
$$= 2\,S(A)(\mathcal{U}_u(\rho)) \le 2\,S(A)(\sigma_{uGu^{\mathsf{T}}}) \le 2\,S(A)(\sigma_{\|G\|_\infty \mathbb{1}_{2N}})$$
$$\le 2\,S_{\mathrm{as}}(A)(\|G\|_\infty \mathbb{1}_{2N}) = 2N_A \ln \frac{e\,\|G\|_\infty}{2} \,. \tag{68}$$

Repeating the same procedure of (68) by switching the subsystems $A$ and $B$ and taking the average with (68) we get

$$I(A;B|R)(\mathcal{U}_u(\tilde{\rho})) \le N \ln \frac{e\,\|G\|_\infty}{2} \,. \tag{69}$$

We have from Proposition 7

$$\frac{1}{2} I(A;B|R)(\mathcal{U}_M(\tilde{\rho})) = \frac{1}{2} I(A;B|R)(\mathcal{U}_T(\mathcal{U}_u(\tilde{\rho})))$$
$$\ge I_{\mathrm{as}}(A;B)(T) - \frac{1}{2} I(A;B|R)(\mathcal{U}_u(\tilde{\rho}))$$
$$\ge S_{\mathrm{as}}(A)(T) + S_{\mathrm{as}}(B)(T) - 2N \ln \frac{e}{2} - N \ln \|G\|_\infty \,. \tag{70}$$

The claim follows by taking the infimum of (70) over $\tilde{\rho}$. $\qquad\square$

**Theorem 2** (Linear growth of squashed entanglement)**.** *For any (generally mixed) initial state $\rho$ with finite covariance matrix and for any time-dependent quadratic Hamiltonian $\hat{H}(t)$ inducing a symplectic evolution $M(t)$ such that the limiting matrix $L(M)$ exists, the squashed entanglement (or CMI entanglement) with respect to the bipartition $AB$ grows asymptotically as*

$$E_{\mathrm{sq}}(\rho(t)) = \Lambda_A\,t + o(t) \quad \text{as} \quad t \to \infty \,, \tag{71}$$

*where the subsystem exponent $\Lambda_A$ is independent of the initial state $\rho$ and can be computed according to Proposition 3.*

*Proof.* Let $\sigma_{\mathbb{1}}$ be the vacuum state of $AB$. On the one hand, we have from Proposition 10 and Corollary 1

$$
\begin{aligned}
E_{\text{sq}}(\rho(t)) &\leq \frac{1}{2} S_{\text{as}}(A)(T(t)^2) + \frac{1}{2} S_{\text{as}}(B)(T(t)^2) + O(1) \\
&= \frac{1}{2} S_2(A)(\mathcal{U}_{T(t)}(\sigma_{\mathbb{1}})) + \frac{1}{2} S_2(B)(\mathcal{U}_{T(t)}(\sigma_{\mathbb{1}})) + O(1) = \Lambda_A t + o(t).
\end{aligned}
\tag{72}
$$

On the other hand, we still have from Proposition 10 and Corollary 1

$$
\begin{aligned}
E_{\text{sq}}(\rho(t)) &\geq S_{\text{as}}(A)(T) + S_{\text{as}}(B)(T) + O(1) \\
&= S_2(A)\left(\mathcal{U}_{\sqrt{T(t)}}(\sigma_{\mathbb{1}})\right) + S_2(B)\left(\mathcal{U}_{\sqrt{T(t)}}(\sigma_{\mathbb{1}})\right) + O(1) = \Lambda_A t + o(t).
\end{aligned}
\tag{73}
$$

The claim follows. $\qquad\square$

## 3 Logarithmic growth

Apart from the linear scaling, it was also shown in [19] that for certain quadratic Hamiltonians, known as metastable, there is also logarithmic contribution to the growth of the entanglement entropy. This result was proven rigorously for Gaussian initial states and time-independent Hamiltonians containing such metastable part, but numerical evidence led to the conjecture that also this behavior is generic for arbitrary and potentially non-Gaussian initial states.

It is therefore a natural question whether we can use the same techniques that allowed us to prove the linear growth for arbitrary initial states to also prove that the logarithmic growth is more general. Unfortunately, the answer is negative. We will show this by providing two counterexamples, where taking the large time-limit and performing the minimization of the right-hand side of (3) do not commute. This does not imply that the conjecture is false, but only that we cannot prove it using the techniques based on the inequality (3).

### 3.1 Classical counterexample

We start with a counterexample in classical probability, which has all the ingredients of the quantum counterexample and is easier to understand. We remind that the Shannon differential entropy [55] of a random variable $Z$ taking values in $\mathbb{R}^N$ is

$$
S(Z) = -\int_{\mathbb{R}^N} p(z) \ln p(z) \, d^N z,
\tag{74}
$$

where $p$ is the probability density of $Z$.

The classical counterpart of a state of a bosonic quantum system is a probability distribution on its phase space. The symplectic form does not play any role in the classical counterexample. Therefore, instead of a bipartite quantum system $AB$ we consider two (generically correlated) random variables $X$ and $Y$ with real values, finite average energy and smooth joint probability density. The counterpart of the time-dependent symplectic transformation $M(t)$ is a time-dependent linear redefinition of $X$ and $Y$. We choose

$$
X(t) = X + t Y, \qquad Y(t) = Y.
\tag{75}
$$

Since in the classical setting there cannot be entanglement, we consider the asymptotic scaling of the mutual information between $X(t)$ and $Y(t)$. On the one hand, for any fixed joint probability distribution of $XY$, such mutual information grows logarithmically with time. Indeed,

we have for $t \to \infty$

$$
\begin{aligned}
I(X + tY; Y) &= S(X + tY) - S(X + tY|Y) = S(X + tY) - S(X|Y) \\
&= S(X/t + Y) + \ln t - S(X|Y) = \ln t + S(Y) - S(X|Y) + o(1),
\end{aligned}
\tag{76}
$$

where we have used that

$$
\lim_{t \to \infty} S(X/t + Y) = S(Y).
\tag{77}
$$

On the other hand, we have for any fixed $t \in \mathbb{R}$

$$
\inf_{XY} (I(X(t); Y(t)) + I(X; Y)) = 0,
\tag{78}
$$

where the infimum is performed over all the joint probability distribution for $XY$ with finite average energy and smooth joint probability density. Indeed, let $X$ and $Y$ be independent Gaussian random variables with variances 1 and $\epsilon^2$, respectively. Then,

$$
\begin{aligned}
I(X + tY; Y) + I(X; Y) &= I(X + tY; Y) = S(X + tY) - S(X|Y) \\
&= S(X + tY) - S(X) = \frac{1}{2} \ln\left(1 + t^2 \epsilon^2\right),
\end{aligned}
\tag{79}
$$

which tends to 0 for $\epsilon \to 0$. Then, performing the infimum over the joint probability distribution of $XY$ before the limit $t \to \infty$ changes the asymptotic scaling of the mutual information.

## 3.2 Quantum counterexample

For our quantum counterexample, we consider a system with two bosonic degrees of freedom and basis $\hat{\xi} = (\hat{\xi}_A, \hat{\xi}_B) \equiv (\hat{q}_1, \hat{p}_1, \hat{q}_2, \hat{p}_2)$ describing the respective subsystems $A$ and $B$. We further consider the quadratic Hamiltonian

$$
\hat{H} = \frac{1}{2} h_{ab} \hat{\xi}^a \hat{\xi}^b = \frac{1}{2} (\hat{p}_1 \hat{q}_2 + \hat{q}_2 \hat{p}_1).
\tag{80}
$$

It is easy to check that this Hamiltonian is metastable (as defined in [19]), as the symplectic generator given by

$$
F = \Omega h \equiv
\left(\begin{array}{cc|cc}
 & 1 & & \\
-1 & & & \\
\hline
 & & & 1 \\
 & & -1 &
\end{array}\right)
\left(\begin{array}{cc|cc}
 & & & 1 \\
 & & 1 & \\
\hline
 & 1 & & \\
1 & & &
\end{array}\right)
=
\left(\begin{array}{cc|cc}
 & & & 1 \\
 & & & \\
\hline
 & & & \\
 & -1 & &
\end{array}\right)
\tag{81}
$$

is clearly nilpotent with $F^2 = 0$. The resulting time-dependent symplectic transformation

$$
M(t) = e^{tF} \equiv
\left(\begin{array}{cc|cc}
1 & & t & \\
 & 1 & & \\
\hline
 & & 1 & \\
-t & & & 1
\end{array}\right)
\tag{82}
$$

will stretch a generic two-dimensional parallelepiped, such that its volume grows linearly in time leading to a logarithmic growth of the entanglement entropy (due to the logarithm in (2)). This is exactly the setup where the conjecture of [19] applies, so we may want to consider the right-hand side of (3) for this choice of $M(t)$. Using the same line of arguments as in the proof of Theorem 1, we can bound the right-hand side of (3) as

$$
\begin{aligned}
\text{RHS} &\leq \inf_{\sigma \text{ Gaussian}} \frac{\ln \det[M(t) G_\sigma M(t)^\mathsf{T}]_A + \ln \det[M(t) G_\sigma M(t)^\mathsf{T}]_B - \ln \det G_\sigma}{2} + 2 \ln \frac{e}{2} \\
&\leq \inf_{a,b,c,d} \ln \sqrt{\frac{(bc + adt^2)(ab + cdt^2)}{ab^2c}} + 2 \ln \frac{e}{2} \leq 2 \ln \frac{e}{2},
\end{aligned}
\tag{83}
$$

where we bounded the infimum over all Gaussian state covariance matrices by restricting to diagonal covariance matrices $G_\sigma \equiv \mathrm{diag}(ab, a/b, cd, c/d)$ with $a \geq 1$, $c \geq 1$ $b > 0$ and $d > 0$, so that the mutual information $I(A;B)(\sigma)$ vanishes. The argument of the logarithm approaches 1 as we take the limit $b \to \infty$. The remaining constant $2\ln\frac{e}{2}$ results from bounding the mutual information by Rényi entropies, *i.e.*,

$$
\begin{aligned}
I(A;B)(\sigma) &= S(A)(\sigma) + S(B)(\sigma) - S(AB)(\sigma) \\
&\leq S_2(A)(\sigma) + \ln\frac{e}{2} + S_2(B)(\sigma) + \ln\frac{e}{2} - S_2(AB)(\sigma),
\end{aligned}
\tag{84}
$$

based on Proposition 1 with $N_A = N_B = 1$.

In summary, we find that we can bound the right-hand side of (3) to be smaller than the constant $N\ln\frac{e}{2}$, which is independent of $t$. We also saw explicitly how the order of limits mattered, *i.e.*, if we could first choose $G_\sigma$ and then take the limit $t \to \infty$, we would find a logarithmic behavior. Unfortunately, the limits do not commute and finding the infimum at fixed $t$ shows that the right-hand side of (3) does not grow logarithmically in time. Consequently, we do not see a straightforward extension of our general proof of linear growth that would also cover a logarithmic contribution. Let us emphasize again that this does not imply that the conjecture of [19] is false, but only that the inequality (3) alone is not sufficient to prove it.

# 4 Applications

In the same way that Gaussian states are an approximation to general semi-classical states, quadratic Hamiltonians are an approximation of weakly interacting Hamiltonians. While our result completely lifts the requirement of the state to be Gaussian (compared to previous proofs), it heavily relies on the Hamiltonian to be quadratic, so that the time evolution can be encoded in the symplectic transformation $M(t)$ of the classical phase space. We will now discuss how our results apply to physical systems with (weakly) interacting Hamiltonians and periodically driven quantum systems, including certain quantum field theory models.

## 4.1 Physical systems with unstable Hamiltonians

The linear growth of the entanglement entropy studied in this manuscript is always due to an exponential squeezing, *i.e.*, due to certain entries of the covariance matrix growing as $e^{\lambda t}$. This also implies an exponential growth of the energy, which certainly will not be sustainable for sufficiently large $t$, as the system will either experience back reactions or higher order terms of the approximate quadratic Hamiltonian will kick in. Either way, the phase of linear growth must always be understood as an intermediate phenomena, which will typically transition to a phase of saturation due to thermalization or equilibration, once the physical model of a quadratic model breaks down.

By approximating time-evolution as linear dynamics under quadratic Hamiltonians, we were therefore able to remove the the phase of saturation and thereby enabled a rigorous treatment of the linear growth phase in the $t \to \infty$ limit. In practice, we will only see this phase if there is separation of scales, *i.e.*, if the time-scale of saturation is larger than the time-scale on which the phase of linear growth happens. While one can always fine-tune the initial state to avoid this separation of scale, there are numerous systems where the phase of linear growth will be relevant.

The simplest application of our result are generic quadratic Hamiltonians $\hat{H} = \frac{1}{2}h_{ab}\hat{\xi}^a\hat{\xi}^b + f_a\hat{\xi}^a$ without any explicit time-dependence. Note, however, that only *unstable* Hamiltonians will give a non-zero production rate $\Lambda_A$, which is equivalent to requiring that the symplectic generator $K^a{}_b = \sum_c \Omega^{ac}h_{cb}$ has some real eigenvalues. The prime example

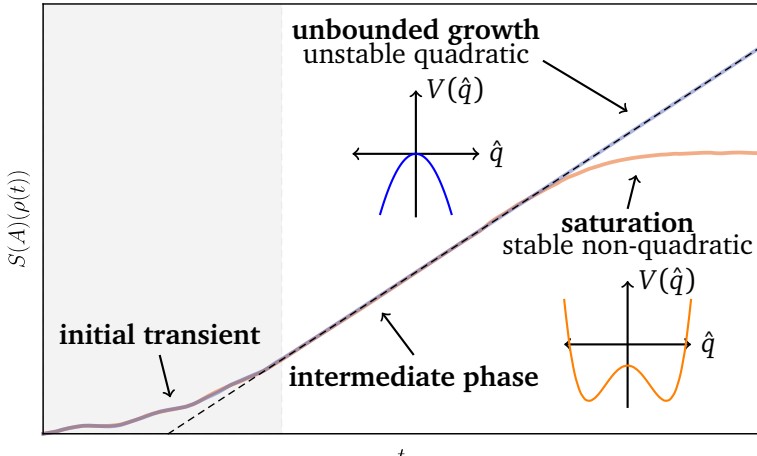

Figure 4: *Illustration: physical Hamiltonians vs. their quadratic approximation.* While an unstable quadratic Hamiltonian leading to unbounded linear growth of the entanglement entropy (in blue) is unphysical, we can still use the predicted linear growth the for intermediate growth phase of a stable non-quadratic Hamiltonian (in orange). We schematically indicate the respective potential $V(\hat{q})$ of a single bosonic mode with Hamiltonian $\hat{H} = \frac{1}{2}\hat{p}^2 + V(\hat{q})$.

of such an unstable quadratic Hamiltonian is the inverted harmonic oscillator $\hat{H} = \frac{1}{2}\hat{p}^2 + V(\hat{q})$ with $V(\hat{q}) = -\frac{1}{2}\hat{q}^2$, which classically corresponds to rolling down an inverted quadratic potential. Such a Hamiltonian is not bounded from below and is therefore typically rendered as unphysical. However, such a quadratic potential can arise as quadratic expansion of a higher order potential, such as $\hat{V}(\hat{q}) = -\hat{q}^2 + \epsilon \hat{q}^4$ with $\epsilon > 0$, that is bounded from below. In order to observe entanglement growth, we require several modes (split into a subsystem $A$ and its complement $B$), which are coupled through at least one unstable mode. As illustrated in Figure 4 and also studied numerically in [19], the asymptotically computed linear growth rate $\Lambda_A$ based on Theorem 1 for the (unphysical) quadratic Hamiltonian can still provide a good prediction for the intermediate phase of linear growth for a physical non-quadratic Hamiltonian. The same to the squashed entanglement $E_{sq}$ based on the results of Theorem 2. More generally, the same reasoning applies to a general bosonic Hamiltonian $\hat{H} = \hat{H}_0 + \epsilon \hat{H}_I$, where $\hat{H}_0$ is at most quadratic (in $\hat{\xi}^a$), while $\hat{H}_I$ is of higher order, as long as $\epsilon$ is sufficiently small.

Unstable quadratic Hamiltonians may also arise as stroboscobic description of time-dependent quadratic Hamiltonians in the context of periodically driven quantum systems. Given a time-dependent quadratic Hamiltonian $\hat{H}(t)$ with periodicity condition $\hat{H}(t + \tau) = \hat{H}(t)$, the unitary time evolution operator

$$U(t) = \mathcal{T}\exp\left(-\mathrm{i}\int_0^t \hat{H}(t')\,dt'\right) \tag{85}$$

satisfies the periodicity condition $U(n\tau + t) = U(t)U(\tau)^n$ for $t \in [0, \tau]$. The long-time asymptotics of the entanglement entropy at times $t = n\tau$ will therefore be governed by the *time-independent* quadratic Hamiltonian $\hat{H}_{\mathrm{strob}} = \frac{1}{\tau}\log U(\tau)$, which is also called the stroboscopic Hamiltonian (as it describes the evolution at discrete snapshots of the system). In this case, $\hat{H}(t)$ can be a perfectly normal physical Hamiltonian that is bounded from below, but the resulting stroboscobic Hamiltonian $\hat{H}_{\mathrm{strob}}$ may turn out to be unstable due to $\hat{H}(t)$ pumping energy into the system. Again, we expect the unstable quadratic approximation to break down

eventually, *e.g.*, when the system starts to back-react or the environment runs out of energy that can be injected into the periodically driven system. Again, the illustration of Figure 4 applies, where we use the long-time asymptotics of the quadratic approximation to understand the intermediate phase of linear growth of a physical model.

The dynamical instability due to a periodic driving of a classical or quantum system is also known as *parametric resonance* [56,57]. The classical evolution can be efficiently described by Floquet theory, where the eigenvalues of the time-dependent symplectic evolution $M(t)$ are approximately $e^{\mu_i t}$. The exponents $\mu_i$ are generally complex and known as *Floquet exponents*, while their real parts correspond to the Lyapunov exponents discussed in section 2, *i.e.*, we have $\lambda_i = \mathrm{Re}(\mu_i)$. Examples of such systems include periodically driven Bose-Einstein condensates [58], the dynamical Casimir effect [59,60] and several cosmological models [61–68].

## 4.2 Quantum field theory subsystems

Strictly speaking, our main theorems do not directly apply to quantum field theories, as we assume a finite number of bosonic degrees of freedom. While the Hilbert space is already infinite-dimensional for a single bosonic mode (just like the quantum harmonic oscillator), we assumed that the phase space has finite dimension. On the contrary, the phase space of a quantum field theory has infinite dimension, and we must therefore ask ourselves whether our analysis still applies.

The key question is what type of subsystems one considers: If one studies local regions of spacetime (such as causal diamonds), the phase space of the associated subsystem will have infinite dimension. Even worse, it is well-known that even free quantum field theories are constructed on a tensor product decomposition over individual modes in momentum space, which cannot directly transformed into a tensor product of local modes. This is captured by the Reeh-Schlieder theorem [69] and also leads to the divergent entanglement entropy found in holographic calculations [70,71]. Our results will not apply to subsystems consisting of *all* degrees of freedom in a local region, as the number of the associated bosonic modes will be infinite.

However, there are still physically interesting subsystems studied in quantum field theory and cosmology that only capture a finite number of bosonic modes. The simplest example are coupled pairs of momentum modes $(\vec{k}, -\vec{k})$ in a spacetime with translational invariance. Such an example was already studied in [18] in the context of inflation and our results on non-Gaussian initial states apply directly, as each pair of modes can be described independently as two bosonic modes becoming entangled.

The other important example where our results can apply is a subsystem of finitely many modes embedded in the field theory. Such a subsystem could represent a detector that couples locally to a certain number (but not all!) of field modes [72,73]. Given a scalar field $\varphi(x)$ with conjugate momentum $\pi(x)$, we can construct a finite number of local modes

$$\hat{q}_i = \int_{\mathbb{R}^3} Q_i(x)\,\hat{\varphi}(x)\,d^3x\,, \quad \hat{p}_j = \int_{\mathbb{R}^3} P_j(x)\,\hat{\pi}(x)\,d^3x\,, \tag{86}$$

where the smearing functions $Q_i, P_j : \mathbb{R}^3 \to \mathbb{R}$ must be chosen such that $[\hat{q}_i, \hat{p}_j] = \mathrm{i}\delta_{ij}$. Here, we assumed that we have chosen a fixed foliation of our spacetime with spatial slices equal to $\mathbb{R}^3$, but the results can be generalized to other cases. Moreover, we can even choose $P_i(x)$ and $Q_i(x)$ to be compactly supported on the spatial slice, so that the subsystem $A$ with phase space spanned by $\hat{\xi}_A^a \equiv (\hat{q}_1, \hat{p}_1, \ldots, \hat{q}_{N_A}, \hat{p}_{N_A})$ is local. While the classical time evolution $M(t)$ will act on the infinite dimensional classical phase space, the subsystem itself will be described by a finite-dimensional phase space, such that the number of modes $N_A$ appearing in Theorem 1 and Theorem 2 will be finite. Such subsystems were studied numerically for Gaussian initial

states in [18] leading to excellent agreement with theoretical predictions and we expect the same for the entanglement entropy and squashed entanglement of non-Gaussian initial states, as described in the present manuscript.

## 5 Summary

The main result of this manuscript is a rigorous proof of the large time asymptotics of the entanglement entropy when a bosonic system is evolved by a quadratic Hamiltonian with instabilities (in the sense of non-vanishing classical Lyapunov exponents). The present work is built upon and heavily relies on a number of previous results, numerical studies and conjectures [16–21] that paved the way for a general proof. Consequently, the result itself does not come as a surprise, but rather concludes the effort of making something rigorous that is quite intuitive: quadratic quantum Hamiltonians are the closest to classical evolution that one can get (as already known by Ehrenfest [74]) and even when one evolves a highly non-classical state, the leading order behavior of the entanglement entropy should be determined by the classical Lyapunov exponents and agree with the Kolmogorov–Sinai entropy rate. Our proof thereby combines classical techniques of characterizing instabilities of Hamiltonian flows and recently discovered properties of the von Neumann entropy. In particular, we demonstrate that the same tools can also be used to prove an analogous result for mixed states.

Our work therefore settles the original conjecture formulated in [16], which was further refined in [17–19]. While it is a natural question if the same tools can also be used to prove that the entanglement entropy grows logarithmically for meta-stable quadratic Hamiltonians and non-Gaussian initial states, as was conjectured in [19], we show that this cannot be easily achieved.

## Acknowledgments

LH thanks Eugenio Bianchi, Ranjan Modak, Marcos Rigol and Nelson Yokomizo for inspiring discussions on the topic during previous collaborations. LH acknowledges support by the Alexander von Humboldt Foundation.

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
