# Peer review of "Linear growth of the entanglement entropy for quadratic Hamiltonians and arbitrary initial states"

_SciPost Physics, doi:SciPost Phys. 12, 021 (2022)_

## Round 1 · Referee Report · Anonymous (Referee 1) · 2021-10-13

Report

The manuscript solves an open problem of the linear scaling of the entanglement entropy of bosonic systems driven by unstable quadratic Hamiltonians with arbitrary bosonic initial state. This scaling has been conjectured earlier in the literature, numerical evidence confirming it have been given, as well as it has been proven for restricted classes of initial states. However, a general rigorous proof which is provided by this manuscript was missing. The breakthrough was possible due to a recent result, ref. 21, (although at the time of writing this ref. 21 was published only in a preprint form) regarding a generalized strong subadditivity of von Neumann entropy for bosonic systems. That new relation allowed for providing a lower bound for the entanglement entropy with non-Gaussian initial states. Together with the upper bound for Gaussian states it was enough to complete the proof. The linear growth is given by means of the so-called Lyapunov exponents.

The paper is rigorous and well written. It provides a detailed introduction and review with quite exhaustive discussion the relevant results from the literature. The main result is formulated as Theorem 1 on page 17. Its proof in a concise and elegant form is based on several Propositions discussed earlier including Propositions which are based on the new strong subadditivity relations and other results from ref. 21.

Authors generalize the result on mixed bosonic states replacing the entanglement entropy by the squashed entanglement. Several consequences and limitations of the applicability of the results for relevant physical systems are discussed.

Summarizing, in my opinion this is an important paper with bold results. I recommend publishing it. Although in my opinion the status of the crucial reference 21 should be clarified first.

---

## Round 1 · Referee Report · Anonymous (Referee 2) · 2021-10-26

Strengths

1- exact results for the squashed entanglement

Weaknesses

no major weaknesses

Report

The authors provide some rigorous results on the growth of the
entanglement entropy in bipartite bosonic systems. The dynamics is
induced by a quadratic unstable bosonic Hamiltonian. In this situation
the authors show that the entanglement grows linearly with time.
The slope of the linear growth is given as sum of Ljapunov exponents.
The result holds also for non gaussian initial states. One of the most
interesting results is that the authors manage to prove the linear growth
of entanglement also for mixed initial states. This is achieved by proving the
linear growth of the squashed entanglement, which is a proper
entanglement measure for mixed states. The pre-factor of the linear growth is the same as that of the entanglement entropy, i.e., it is a sum of Ljapunov exponents.

This paper contains very interesting results. In particular, the results for
the dynamics of entanglement starting from mixed initial states are of great relevance. One reason is that the authors provide some of the few exact results available for the squashed entanglement, which is an interesting entanglement measure, although very difficult
to compute. The paper is well written and the results are clearly presented.
For these reasons I definitely recommend its publication in Scipost.

---

## Editorial Decision

published